# Self-driving lab discovers principles for steering spontaneous emission beyond conventional Fourier optics

Saaketh Desai, Sadhvikas Addamane, Jeffrey Y. Tsao [ORCID], Igal Brener [ORCID], Remi Dingreville [ORCID] & Prasad P. Iyer [ORCID] [✉]

We develop an autonomous experimentation platform to accelerate interpretable scientific discovery in ultrafast nanophotonics, targeting a novel method to steer spontaneous emission from reconfigurable semiconductor metasurfaces. Despite the potential of reconfigurable semiconductor metasurfaces with embedded sources for spatiotemporal control, achieving arbitrary far-field control remains challenging. Here, we present a self-driving lab (SDL) platform that addresses this challenge by discovering the governing equations for predicting the far-field emission profile from light-emitting metasurfaces. We discover that both the spatial gradient (grating-like) and the curvature (lens-like) of the local refractive index are key factors in steering spontaneous emission. The SDL employs a machine-learning framework comprising: (1) a variational autoencoder for generating complex spatial refractive index profiles, (2) an active learning agent for guiding experiments with real-time closed-loop feedback, and (3) a neural network-based equation learner to uncover structure-property relationships. The SDL demonstrates up to a four-fold enhancement in peak emission directivity (up to 77%) over a 74° field of view within ~300 experiments. Our findings reveal that combinations of positive gratings and lenses are as effective as negative lenses and gratings for all emission angles, offering a novel strategy for controlling spontaneous emission beyond conventional Fourier optics.

Self-driving labs (SDLs) represent a transformative approach to scientific discovery, employing machine-learning (ML) models to autonomously conduct experiments[1–10]. Current experiments within SDLs focus on low-dimensional problems due to the challenges in interpreting high-dimensional ($>10^3$ degrees of freedom) data structures[9,11–17]. Hence, SDLs have primarily concentrated on optimization tasks within low-dimensional or well-understood search spaces, accelerating discovery in material science and chemistry[18–25]. However, realizing interpretable scientific discovery[26] for high-dimensional problems presents a significant challenge, as it involves navigating unknown high-dimensional spaces to establish new verifiable facts or concepts. Additionally, high-throughput automation of experiments

(closed-loop) is necessary for tackling high-dimensional problems, which additionally limits SDLs[27,28]. ML models excel at learning correlations in high-dimensional spaces but struggle with extrapolation and interpretation, especially in the physical sciences, where they often act as "black-boxes", failing to learn the underlying physical principles[29–32]. This inherently limits the generalizability of ML models since researchers cannot explain 'why' a particular discovery makes sense, for instance, in the form of an equation representing the process[33,34]. Since the advancement of scientific research over the past century has been successful in realizing interpretable solutions by following the scientific method[35–37], we hypothesize that an ML framework implementing the scientific method can realize interpretable discovery. We

Center for Integrated Nanotechnologies, Sandia National Laboratories, Albuquerque, NM, USA. [✉]e-mail: ppadma@sandia.gov

therefore envision an SDL to generate high-dimensional experiments, select optimal hypothesis-driven experiments for testing, identify features of relevant optimal experiments, and uncover the relationship between these features and experimental results in an interpretable form. Here, we address two objectives: a) develop a machine-learning framework for autonomous scientific discovery, and b) apply this framework to discover a novel approach to steer spontaneous emission.

To achieve these objectives, we develop an ML framework for autonomous scientific discovery in three steps - specifically to address the needs and limitations of current SDLs:

a)  High dimensionality of inputs: We leverage the manifold hypothesis[38,39] within the physical sciences, which suggests that a complex physical system requiring a large number of parameters at first glance, can be described with far fewer independent parameters. We employ generative models, specifically variational autoencoders (VAEs)[40], to generate high-dimensional experiments beyond state-of-the-art, from a low-dimensional continuous latent space.

b)  Cost of Experiments: Active learning (AL)[11] then selects optimal experiments from the VAE's (low-dimensional) latent space, to develop an efficient design of experiments overcoming limitations in exploring high-dimensional spaces. Specifically, active learning predicts the next experiment to be conducted, balancing exploration and exploitation of the input space with appropriate acquisition functions.

c)  Interpretability of Results: Understanding experimental results is crucial, yet generative and active learning models often lack human interpretability. To bridge this gap, we develop a neural network-based equation learners (nn-EQLs) that uncover interpretable equations[26,41]. Our approach combines the expressive power of neural networks with physics-driven intuition to learn interpretable structure-property relationships through closed-loop experimental feedback.

We apply our ML framework to the problem of steering spontaneous emission, a challenging task with significant potential for clean

energy solutions. Spontaneous light emission, as seen in light-emitting diodes (LEDs) and thermal lamps, lacks spatio-temporal control, but achieving such control could revolutionize fields like remote sensing and holographic displays[42–48]. Traditional methods for controlling coherent (e.g., phased array optics[49] for lasers) light are not compatible with spontaneous emission. Light-emitting metasurfaces, composed of sub-wavelength periodic arrays of optical resonators with embedded emitters, offer a novel way to control spontaneous emission, and have demonstrated reconfigurable control of spontaneous emission through spatially periodic refractive index modulation[50–56]. However, predicting and controlling emission patterns from aperiodic refractive index modulation remains challenging due to a lack of suitable models and simulation tools (see Supplementary Information Section S1). Our approach involves utilizing the ultrafast (<1 ps) reconfigurability (through refractive index modulation with optical free-carrier injection) of the metasurface to realize nearly arbitrary phase-array optical elements, mimicking arbitrary high-dimensional spatial index profiles. Our SDL leverages the degrees of freedom enabled by the reconfigurability of the metasurfaces to discover the relationship between spatial refractive index profiles and emission patterns through closed-loop, noisy experimental feedback. Based on the exploration results of the AL agent within the latent space of the VAE, we improved the peak directivity by of an order of magnitude realizing up to 67% over a wide field of view (80°) when compared with state-of-the-art devices[56] and realized human interpretable equations describing the steering process (Fig. 1a). The efficacy of combining generative models with active learning to tackle high-dimensional spaces demonstrates the acceleration of autonomous experimentation platforms towards scientific discovery. Using the results from our self-driving lab, we discovered a novel approach to steer incoherent emission—going beyond conventional Fourier optical principles that are based solely on momentum matching of light. We demonstrate that a spatial index profile that is a combination of a positive (convex) lens and a positive (sawtooth) grating, when imposed on the metasurface, can steer light emission across a 74° field of view. Similarly, a negative (concave) lens and a negative (sawtooth) grating also steer emission across the same field of view, while other spatial index

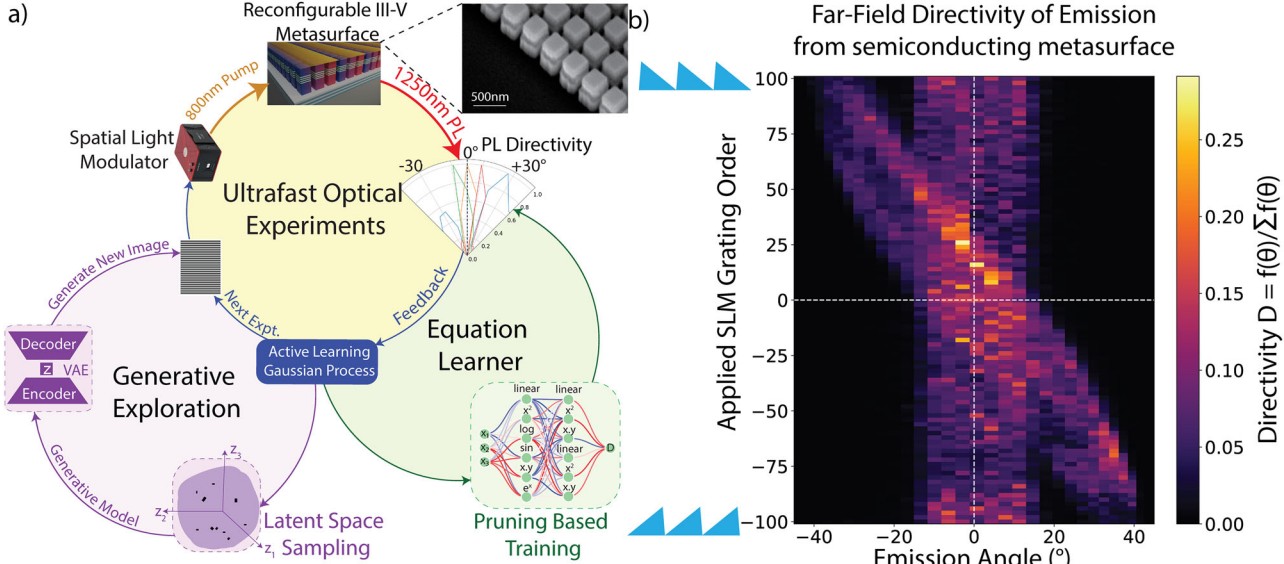

**Fig. 1 | Spontaneous Emission Steering. a** Self-driving lab framework: an active learning agent drives a generative model (variational autoencoder, VAE) and the ultrafast optical experiment. A 40 fs pulsed pump laser at 800 nm images a spatial light modulator onto a reconfigurable metasurface (scanning electron microscope image at the top right). This setup measures the directivity of photoluminescence (PL) from the metasurface with closed-loop feedback. The latent space discovered

by the active learning surrogate model is recast as a human-interpretable equation using a neural network-based equation learner. **b** Far-field directivity emission from the metasurface under saw-tooth shaped (uniform spatial momentum) pump patterns applied on the SLM, with the grating order varying from −100 to +100. The directivity (D) of emission is defined as the ratio of the signal towards a given angle ($f(\theta_i)$) to the sum of signals over all angles ($\Sigma_j f(\theta_j)$).

combinations (e.g., positive (negative) lens with a negative (positive) grating) do not steer the emission from our reconfigurable light-emitting metasurface. Through our approach, we provide a potential pathway for augmenting scientific intuition using neural network equation learners, taking a step beyond scientific discovery to understanding the principles governing spontaneous emission.

## Results and discussion

To rapidly measure spontaneous emission, we develop an automated, closed-loop, ultrafast momentum-resolved photoluminescence (PL) measurement setup featuring a reconfigurable GaAs metasurface with embedded, light-emitting, InAs quantum dots, grown epitaxially on a reflective distributed Bragg reflector. The local intensity of the ultra-fast optical pump (800 nm, 40 fs pulse-width laser, 2–3 mJ/cm$^2$ at 1 kHz) dynamically changes the local refractive index of the GaAs metasurface resonators through free-carrier refraction (Fig. 1a). An active learning (AL) agent drives the experiments, sampling the smooth lower-dimensional latent space of a variational autoencoder (VAE) to generate new optical pump patterns. These patterns are projected onto the metasurface as intensity patterns using a spatial light modulator (SLM) to realize the spatial refractive index profile on the metasurface. The far-field directivity of the PL is captured using a lock-in detector scanning the back focal plane of the metasurface emitter (see Supplementary Information S1). For this SDL exemplar, the optical pump pattern setting up the spatial index of profile on the metasurface forms the input image (generated by the VAE) for the experiment and the far-field intensity distribution of metasurface emission forms the output, which is used by the AL agent to generate the next experiment.

The GaAs metasurface is designed to achieve nearly a 0–2π phase shift in reflection as a function of local optical pump intensity and is fabricated using electron beam lithography and dry etching as described previously in other work[51,56]. The metasurface shows overlapping peaks in reflection and PL spectra (Fig. S1), indicating that optical resonances (peaks in spectra) measured in reflection can enhance the far-field emission from the metasurface. We demonstrate that the metasurface steers light emission (Fig. 1b) over an 80° field of view under one-dimensional uniform momentum profiles created by saw-tooth patterns with different spatial frequencies on the optical pump structured using the SLM. Far-field emission directivity measurements (D $= \frac{f(\theta_i)}{\sum_j f(\theta_j)}$, where f($\theta_i$) is the steered signal towards angle $\theta_i$ reveal a band of emission between ±14° (due to the distributed Bragg reflector substrate) for all applied grating orders, with some emission steered to off-normal angles responding to the pump pattern's spatial momentum. This result (Fig. 1b) indicates that only part of the metasurface emission follows the known momentum matching principles of spontaneous light emission steering[50,51,56–63]. Therefore, ML has the potential to discover better solutions beyond momentum matching principles, solving the inverse problem of predicting the optimal spatial refractive index profile to maximize directivity towards a given angle. In our work, the ML framework controls the closed-loop ultrafast optical experiment to maximize the metasurface emission directivity towards a desired angle.

We first benchmark the components of our ML framework to ensure that the VAE can generate a wide variety of patterns beyond human intuition[40] and that the AL agent can search over known spaces to rediscover known results. We quantify the generative capability of the VAE by visualizing the local-slope distribution of the optical pump patterns generated by the VAE, demonstrating that the generated optical pump patterns exceed the state-of-the-art (training set) by two orders of magnitude (See "Methods" and Supplementary Information S2). We also illustrate that the traditional grating order-based patterns and polynomial patterns lie on a subset of the space of patterns generated by sampling the VAE's latent space (See

Supplementary Information S2). This analysis confirms the validity of the manifold hypothesis in our problem, such that sampling this low-dimensional space is sufficient to discover governing equations relating features of high-dimensional pump patterns to the observed directivity. The AL agent begins by using a limited initial training set (optical pump patterns and their associated directivity) to predict directivity across various optical pump patterns. An acquisition function then identifies patterns that could maximize directivity. The ultrafast, automated PL experiment is then performed by projecting this pattern on the metasurface, and the measured result (and associated noise, statistics) is added to the training set of the AL agent. This loop continues until an optimum is reached, or a pre-set experimental budget is reached. Here, the active learning agent is limited by our experimental budget, and we thus note that maxima in directivity (figure of merit) achieved represent local maxima, and not a global maximum of the optimization problem. We find that the AL agent, when searching over the space of possible grating orders, re-discovers the optimal grating order for steering into a given direction with an order of magnitude fewer number of experiments than brute-force sampling (See "Methods" and Supplementary Information S2). Given the success of the AL agent in finding pump patterns described using a single parameter (grating order), we now use AL to find pump patterns described using a complex set of low-dimensional features, i.e., the VAE latent space.

In Fig. 2a, we show the results of the AL agent maximizing the steered signal towards two different emission angles without prior knowledge (i.e., the AL agent has no prior information of the VAE, the experimental noise, and past results). The AL agent samples the VAE's latent space, aiming to maximize directivity while trying to minimize the number of experiments. Each experiment (dot in Fig. 2a) starts with the AL agent predicting a point in the VAE's latent space representing a potentially high-directivity curve. This point generates a one-dimensional curve (Y$_{VAE}$ of length 3840) through the VAE decoder, which is transformed into an optical pump pattern in two steps. i) phase wrapping and normalizing the curve to the 8-bit resolution of the SLM to generate Y$_{SLM}$ = (Y$_{VAE}$%2π)/2π,; and ii) repeating Y$_{SLM}$ along the orthogonal axis (2160 times) to form a two-dimensional image (3840 × 2160 pixels) projected onto the metasurface. While the VAE could generate patterns with high-spatial frequency going beyond the diffraction limit of the optical system, we note that the experimental setup imaging the pump pattern from the SLM to the metasurface will naturally smooth out these high-frequency elements. The noise in the far-field emission has four independent sources: the ultrafast-pulsed laser, SLM, infrared detector, and lock-in amplifier. The directivity of PL is estimated with 10 repeats of the experiment to derive the mean (output) and its standard deviation (noise, see Supplementary information S1). Combining the VAE's enhanced generative capability with the AL agent's efficiency, we maximized the directivity of spontaneous emission from the GaAs metasurface.

We find that the AL agent sampling the VAE's latent space improves peak directivity across a 74° field-of-view within 300 closed-loop iterations without prior knowledge (Fig. 2a). Compared to the state-of-the-art saw-tooth grating patterns[56], the absolute directivity of emission increases by an average of 2.2x (Fig. 2b), with a peak improvement of 3.77x at 14.4°. Here, we note that the AL agent utilizes the real-time noise (red error bars in Fig. 2a) in each experiment (See "methods") performed by the SDL. Notably, the absolute directivity peak of 67% is one of the highest reported for classical static LEDs, which typically require bulky reflector lenses and collimators to achieve similar directivity[64]. We demonstrate that a dynamically reconfigurable light-emitting metasurface can be designed to increase emission directivity based on an aperiodic spatial refractive index pattern imposed on the emitter. The patterns discovered by the AL agent, resulting in high directivity, represent spatial index profiles beyond conventional optical elements, which are typically defined

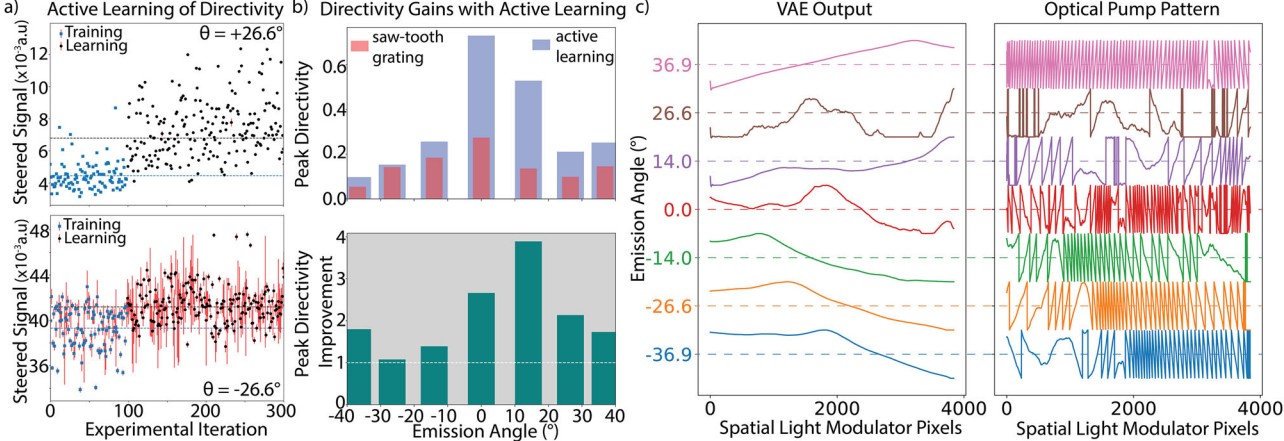

**Fig. 2 | Improving Emission Directivity with Active Learning. a** Top (bottom) panel shows the active learning result for the emission angle at +(−) 26.6°. Blue dots represent training points (Sobol sampling) in the VAE latent space, and black points represent improved steering signals learned by the active learning agent. The blue dashed line indicates the average of the training points, while the black line indicates the average of the learning points. Red vertical stripes show measurement errors averaged over 10 repeats. **b** Relative improvement in peak directivity by the active learning agent across multiple far-field emission angles through closed-loop

experimental feedback. The top panel shows the peak directivity of emission optimized for each far-field angle, with the active learning agent's values in blue and the saw-tooth grating values in red. The bottom panel shows the relative improvement in peak directivity for different emission angles enabled by the active learning agent. **c** Left panel shows the optimal VAE output (normalized) learned by the active learning agent, and the right panel shows the normalized 1–D pump pattern transforming the VAE output for loading into the SLM: $Y_{SLM} = (Y_{VAE} \% 2\pi)/2\pi$ for different emission angles.

based on symmetries governed by the spatial phase profile of the device. Additionally, we note the patterns discovered by the AL agent do not have any high-spatial frequency features beyond the diffraction limit of the system. Thus, the AL agent, leveraging the VAE's generative potential, provides a novel approach to maximize emission directivity, surpassing state-of-the-art methods to achieve comparable performance to commercial LEDs[64] without additional packaging or bulky optics, while retaining the ability to steer emission over 74°.

Remarkably, we discover that the optimal VAE outputs for different emission angles, as identified by the AL agent, can be described as a combination of the spatial phase profiles of a lens and a grating (Fig. 2d). The spatial (x - pixels on SLM) phase (y) profile of an optical lens (used to focus light) is a parabola: $y = ax^2$ (where a represents the lens curvature), while the saw-tooth gratings (used to deflect light) have a linear phase profile: $y = bx$ (where b defines the deflection angle, with b = grating order/3840 pixels). Thus, the VAE patterns discovered by the AL agent can be described as $Y_{VAE} = ax^2 + bx$, and the final optical pump pattern on the SLM becomes $Y_{SLM} = (ax^2 + bx) \% 2\pi / 2\pi$. This finding, enabled by the AL agent exploiting the generative capability of the VAE, surpasses human intuition, which typically relies on momentum matching (or Fourier transform-based) principles. Current methods for steering light depend solely on grating orders (bx) or the linear spatial gradient established by the refractive index or size profile of the metasurface resonators. Here, for the first time, we discover a fundamentally new way to steer light from light-emitting metasurfaces with high directivity through the AL agent.

To formalize our visual insights, we quantify the statistical correlation between the VAE's latent space ($z_{1–4}$) and the physical properties of the SLM pump pattern ($Y_{SLM}$): **a**- spatial curvature ($\partial^2 y_{vae}/\partial x^2$), **b**- spatial gradient ($\partial y_{vae}/\partial x$), **A**- average pump intensity (<$Y_{SLM}$>), and **ω**- largest spatial frequency (|FourierTransform{$Y_{SLM}$}|$_{max}$). These features are commonly used to describe the spatial refractive index profile of the optical pump pattern. Spearman correlation coefficients[65], averaged over 10,000 VAE-generated curves (using Sobol Sampling), reveal: a) $z_4$ weakly correlates with the spatial curvature, **a**; b) $z_3$ negatively correlates with the average pump intensity, **A**; and c) the VAE's latent space dimensions are orthogonal (See "Methods"). The rest of the latent space shows no or weak correlation with other physical propertie of the pump pattern. While Spearman

correlations indicate isolated correlations, Sobol' sensitivity indices[66–68] help us understand the combined correlations of multiple latent space variables. These indices show that no single latent space dimension correlates strongly with a, but combinations of dimensions do. For spatial gradients, only $z_4$ correlates weakly in isolation, but a combination of $z_{1–3}$ and $z_4$ correlates strongly with **b**. This indicates that the AL agent discovered a correlated sub-space of patterns with high performance across all angles, which are interpretable and tied to physically relevant quantities. We experimentally validate the AL agent's discovery using a parameter sweep on the optical pump pattern (Fig. 3c–e, Supplementary Information S3), finding that specific combinations of lens and grating pump patterns result in high directivity. Combining lens and grating pump patterns creates an aperiodic spatial refractive index profile, dynamically reconfiguring the metasurface to achieve high directivity. Our work thus reveals a new structure-property relationship governing spontaneous emission steering at the nanoscale, relating aperiodic spatial refractive index (momentum) profiles and directivity beyond current momentum matching principles.

We translate the structure-property relationship discovered by the AL agent and the VAE to a human-interpretable equation describing the directivity (D) of emission as a function of the latent space (a, b) using a neural network-based equation learner (nn-EQL)[26,41]. The nn-EQL is a two-layer neural network with non-linear activation functions (e.g., addition, $t^2$, sin(t), cos(t), multiplication). We train the nn-EQL to minimize mean-squared error through backpropagation, while using iterative pruning (>90% sparsity)[69], to obtain Eq. (1), see Fig. 4. Details on the nn-EQL setup, and the pruning process are described in "Methods".

$$D_g(0^o) = 0.18(a - 0.033b)^2 + 0.029b^2 + 0.04\sin(3.52a - 0.04b) + 0.65 \quad (1)$$

The pruned network is read out as an equation and further simplified using Python packages (e.g., sympy). The nn-EQL distilled equation captures the oscillatory behavior observed in the dataset with a 'sin(3.52a−0.04b)' term, while '0.18(a−0.033b)²' describes the dependence on spatial curvature. (See Fig. 4a, "Methods") This equation is a subset of a master equation describing spontaneous emission

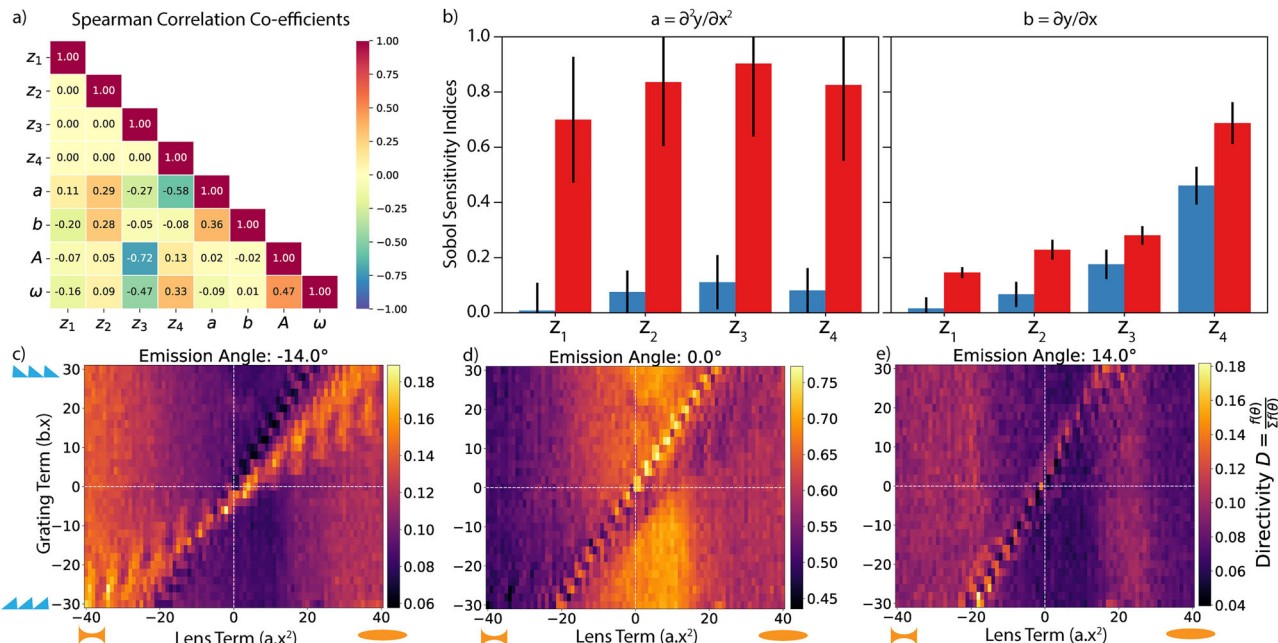

**Fig. 3 | Latent space discovery with active learning. a** Spearman correlation coefficient among latent dimensions learnt by the VAE, a,b-spatial curvature and gradient of the pump, A,ω-average intensity and the highest spatial frequency of the SLM pump profile. **b** Sobol correlations between latent space dimensions of the VAE with the spatial curvature (left panel), and the gradient (right panel) of the optical pump profile. The blue bars indicate correlation with respect to an

individual variable ($z_i$) while the red bars highlight the correlation in the presence of other latent space variables. The black stripes on top of the bars indicate the possible error bar in the correlation statistic. **c–e** Measured directivity of emission from the metasurface as a function of spatial curvature, a and gradient, b for steering results −14°,0,and +14° respectively.

steering towards 0° concerning the spatial gradient and curvature of the refractive index profile. Repeating this process for other steering orders allows us to develop a generalizable phenomenological model describing light emission from the metasurface. Examples of such additional equations are provided in the Supplementary Information Section S3. Analyzing these equations, we can observe a consistent trend wherein the optimal a/b ratio for maximizing light emission gradually increases as the redirection angle changes from −36° to +36°.

The spatial refractive index profile, representing a combination of a lens and a grating, presents an intriguing perspective from Fourier optics. Current approaches to steer spontaneous emission rely on momentum-matching principles, but our results (Fig. 3) and the obtained equation (Fig. 4) suggest a new operational principle. This principle considers not only momentum matching (using grating orders, **b**) but also the gradient of momentum (spatial curvature of the index, **a**), akin to lens-like characteristics. For instance, in Fig. 3c, combining positive lens (a) and positive grating (b) values steers light effectively (high directivity) in the first quadrant. However, combining a positive lens with a negative grating (fourth quadrant) shows no steering. In classical optics, a positive lens collimates an LED source, followed by a grating to steer the emission. The ML framework enables combining the lens and grating into a single optic on the light-emitter itself. Negative grating orders combined with positive lens characteristics (fourth quadrant) completely remove the capacity to steer light for all angles, contrary to classical optics. The second and third quadrants of Fig. 3c present an even more intriguing scenario where a negative (concave) lens with embedded emitters can steer light in the far-field with an additional negative grating order. A negative lens typically increases the divergence of incident light, counterintuitive to collimating a diverging source unless the source was initially converging. Multiple sets of light emission pathways trapped in the substrate may be out-coupled into free space along the measurement direction, but further modeling of the system is necessary to verify this

possibility. The equivalence in performance between positive and negative lenses (with their corresponding gratings) for steering light emission in the far-field suggests that observed spontaneous light-matter interactions surpass classical Fourier optics' understanding based only on momentum matching.

In summary, we demonstrate a self-driving nanophotonics lab capable of uncovering and elucidating novel structure-property relationships in nanoscale light-matter interactions. Our approach integrates a generative model (VAE), an active learning agent, and an equation learner network, constituting the core elements of a self-driving lab. Leveraging the VAE's capacity to generate patterns beyond human intuition from a condensed design space representation, the active learning agent efficiently optimizes for the metasurface's operational property (directivity of emission) using closed-loop feedback to minimize experimental iterations. The patterns unearthed by the active learning agent across multiple emission angles revealed new latent space variables (such as spatial curvature of the refractive index), enhancing control over spontaneous emission directivity. The nn-EQL distills the active learning agent's discovery into concise equations, offering much-needed human interpretability to machine-learning models. The active learning agent identified optical pump patterns from the VAE, steering spontaneous emission from resonant metasurfaces 2.2x more effectively than human-intuition-driven saw-tooth pump patterns over a 74° field of view, peaking at 67% at normal incidence. We discovered that a spatial index profile formed by combining a lens and a grating outperforms a grating profile alone in steering spontaneous light emission. Specifically, the agent unveiled that these optical components (lens and grating) may be amalgamated into a single phase-space optical pump pattern on a metasurface, transcending current understanding based on Fourier optics. Moreover, we discover a concise structure-property relationship linking the spatial refractive index profile to emission directivity as an equation, facilitating the realization of energy-efficient spontaneous light sources (such as LEDs, thermal lamps, etc.). We anticipate that the

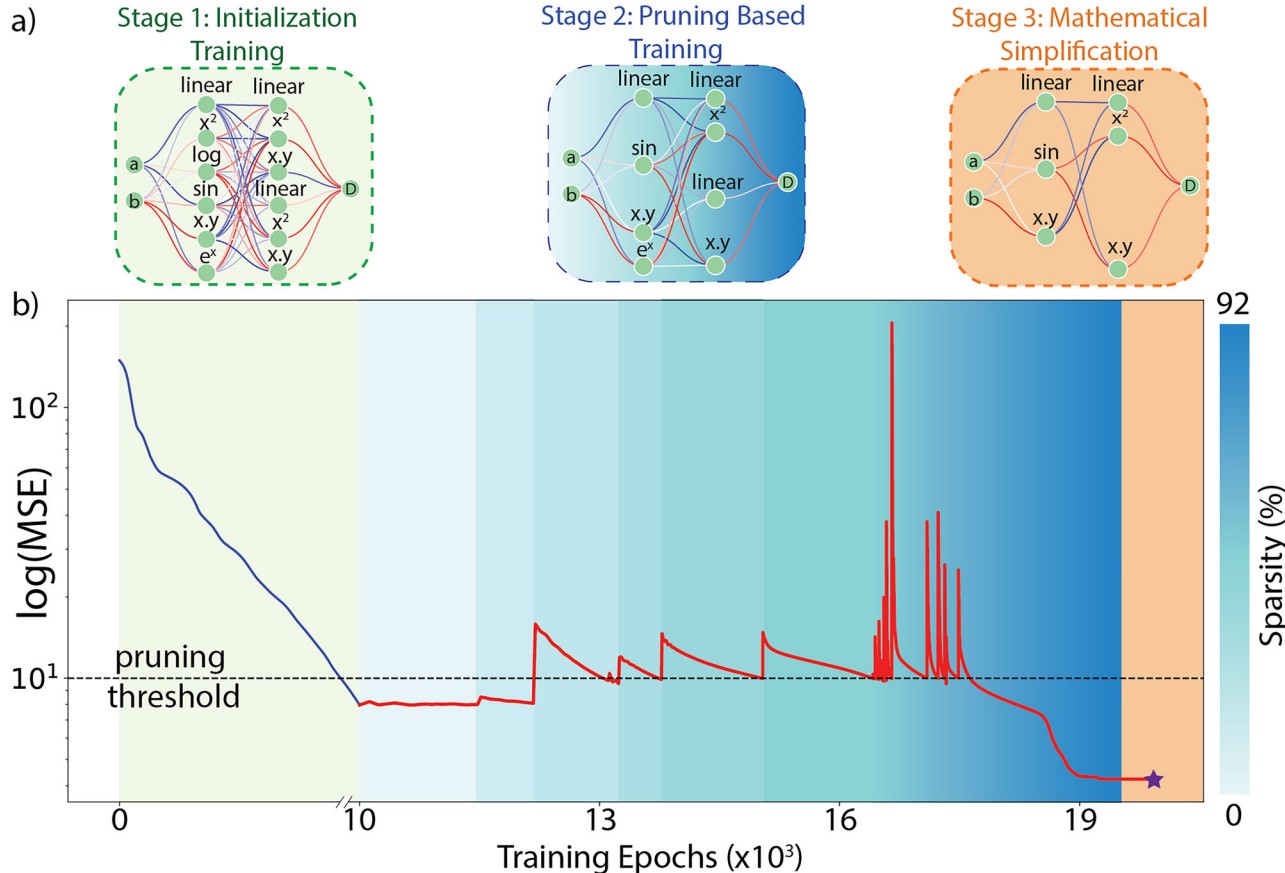

**Fig. 4 | Equation learner framework. a** Our equation learner framework uses a customized neural network with physics-based activation functions defined for each neuron. Stage 1 first performs an initial fit to the dataset, establishing an acceptable error level on the training and validation set (green). Stage 2 then iteratively prunes this network, removing neurons that have the lowest contribution in each layer, which is repeated until the highest level (e.g., 90%) of sparsity before the loss terms increase. Finally, in Stage 3, we write an equation using the neural network's weights and activation functions and simplify it using sympy[75]. **b** The training process (logarithm of the mean squared error of the network vs the training epochs) for generating equations represents the three stages described in Fig. 4a. The dashed horizontal represents the loss-threshold for initiating the pruning of the least contributing neurons the network. The color bar represents the sparsity progression during the process.

methodologies and outcomes demonstrated herein to establish a self-driving ultrafast nanophotonics lab could be generalizable to other physical sciences. This approach has the potential to overcome limitations of human intuition and theoretical frameworks, leveraging generative machine-learning frameworks to drive new scientific discoveries.

## Methods

### Variational autoencoder (VAE)

We train a VAE to generate arbitrary pump patterns, represented as one-dimensional curves of length 3840 pixels. Given a pump pattern $X$, a VAE encodes this pump pattern into a low-dimensional latent space representation $Z$, using an encoder network that learns the distribution $Q(Z|X)$. In this work, we use a latent space representation of size four, making the latent space a four-dimensional Gaussian distribution. During training, the accuracy of the learnt latent representation $Z$ is determined in part by the ability of the VAE to reconstruct the input pump pattern $X$ from the latent space representation $Z$. The VAE achieves this reconstruction by using a decoder network that learns $P(X|Z)$. In this work, we use six fully connected layers for the encoder and the decoder. The objective of the VAE during training is to minimize Evidence Lower Bound (ELBO) loss:

$$L_{VAE} = ||\hat{X} - X||_2^2 + KL(Q(Z|X)||p(Z)) \qquad (2)$$

Here, the first term is the L2-norm between the set of reconstructed pump patterns $\hat{X}$, and the set of ground truth patterns $X$. The second term is a KL-divergence that measures the difference between the encoder-learned distribution $Q(Z|X)$, and the prior distribution $p(Z)$, assumed here to be $N(0, I)$. Once the VAE is trained, new pump patterns are generated by sampling the learned latent space of the VAE. The generated pump patterns are then transformed into two-dimensional images by repeating the intensity of the pump pattern along the y-axis. These images are projected onto the SLM for evaluation of spontaneous emission steering. The training set for the VAE encompasses both grating-order-based periodic patterns and aperiodic patterns comprising multiple frequencies and linear/quadratic curves. In total, 50,000 one-dimensional pump patterns are used for VAE training, using the Pytorch package[70] and the Adam optimizer[71] for training. See Supplementary Information Section S2 for benchmarking of the generative capability of our trained VAE.

### Active learning on the latent space of the generative model

We search for optimal pump patterns by navigating the latent space of the trained VAE using active learning. Each point in the latent space represents a pump pattern (obtained using the VAE's decoder network), and active learning efficiently searches the latent space (i.e., the space of pump patterns that could be generated by the VAE) to find optimal pump patterns. We define optimal pump patterns as patterns

with high directivity $D = \frac{f(\theta_i)}{\sum_j (f(\theta_j))}$, i.e., patterns that steer emission maximally in a desired direction $\theta_i$, while minimally steering emission to other angles. Active learning begins with an initial dataset of pump patterns and associated directivity ($D$) values. This set of pump patterns is chosen by choosing points in the latent space using Sobol's sampling and using the VAE's decoder to obtain pump patterns. Directivity is measured using the automated experimental setup (see Supplementary Information), along with uncertainty. We assume the directivity measurement is assumed to contain errors from the pump power (combining modulation and laser measurement) $\Delta P$, and the thermal noise of the detector $\Delta S$. The uncertainty in the directivity is thus calculated as $\frac{\Delta D}{D} = \sqrt{\left(\frac{\Delta P}{P}\right)^2 + \left(\frac{\Delta S}{S}\right)^2}$. Using this initial dataset, a Gaussian process model GP predicts directivity, with uncertainty, across the latent space, i.e., $D(z) = GP(\mu(z), K(z, z'))$; where $z$ is a point in the latent space of the VAE, $\mu(z)$ is the average directivity for the pump pattern represented in the VAE dimension as $z$, and $K(z, z')$ is a kernel function representing covariance in directivity between two pump patterns represented as latent space points $z$ and $z'$. See[72] for more details on Gaussian process models. The directivity prediction from the Gaussian process model, along with uncertainty, is used for determining the next experiment that will be conducted. The next experiment $z^*$ is determined by an acquisition function, such as Expected Improvement (EI), with the intention of balancing exploration and exploitation in the latent space. Specifically, the next experiment with the EI acquisition function is chosen as: $z^* = argmax\ E(max(GP(z) - GP(z^{curr})), 0)$, where $z^{curr}$ is the point in the latent space with the best directivity so far, and $z$ is any point in the latent space. The EI acquisition function thus chooses the next experiment to be conducted at a latent space $z^*$ where the Gaussian process model predicts the highest improvement in directivity, compared to the best point predicted so far. In this work, we use the Ax package[73] for the active learning, using 100 points in the latent space as the initial dataset (sampled using Sobol' sampling[74]). Subsequently, 1000 experiments are conducted using the EI acquisition to find optimal pump patterns.

### Analyzing correlations in latent space
Correlations in latent space are analyzed using Spearman correlations and Sobol' sensitivity indices. The Spearman correlation is defined as: $\rho_{a,b} = \frac{cov(r_a, r_b)}{var(r_b)var(r_b)}$ where $r_i$ is the rank of variable $i$ (highest value is rank 1). Spearman correlations range between $-1$ and $1$ and indicate correlations between pairs of variables. Moving beyond pairs of variables in isolation, Sobol' sensitivity indices indicate the effect of sets of variables on a quantity of interest. Sobol' sensitivity indices are of two types: first-order sensitivity indices and total-order sensitivity indices.

First-order sensitivity indices are defined as: $S_{1,i} = \frac{Var_{X_i}\left(E_{X \neq X_i}(Y|X_i)\right)}{Var(Y)}$ and measure the variance in the quantity of interest $Y$ when varying one variable $X_i$, and averaging over all other variables. Total-order sensitivity indices are defined as: $S_{T,i} = \frac{E_{X_{\sim i}}\left(Var_{X_i}(Y|X_{\sim i})\right)}{Var(Y)}$ and are the variance in the quantity of interest as combinations of $\{X_i\}$ are varied.

### Equation learner network
To distill active learning experiments into an interpretable form, we employ a custom equation learner network (EQL). The EQL is formulated as a dense feed-forward neural network (we use Pytorch[70]) with custom activation functions applied to each neuron. These custom activation functions are inspired by terms present in equations in the physical sciences (sin, cos, exp, product, etc.). The EQL is trained in three stages: (1) A two-layer network is trained to achieve an accurate fit to the data (without overfitting), (2) The trained network is pruned

to a smaller network that achieves similar accuracy to the network trained in stage 1. We prune the EQL by removing weights with the least 'contribution' in each layer, followed by re-training for a few epochs to allow other weights to adjust to the removal of weights with the least contribution. Contribution here is defined as the product of the weight value and the value of the neuron activation from the previous layer. We use contribution as a metric instead of conventional magnitude-based pruning approaches[69] to account for non-custom activation functions that are not monotonic with weight values (e.g., cos) (3). The pruned network is read out in terms of a human-readable equation using packages such as Sympy[75].

### Coupling experiments with powerful computing platforms
Our self-driving lab is driven by a laboratory computer capable of instrument control via a Python API. To overcome computational limitations, active learning and neural network equation learning is performed on a local machine with four Tesla V100 GPUs, while experimental instruments are controlled via a laboratory computer.

## Data availability
The experimental beam steering data generated in this study have been deposited in the Zenodo database [https://doi.org/10.5281/zenodo.17253431] under the Creative Commons Attribution 4.0 International license. Also included is a Jupyter notebook to visualize the raw data and simple properties of the pump patterns.

## Code availability
The code is available at https://github.com/saakethdesai/autoscilab.

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

## Acknowledgments

This work was supported by the US Department of Energy (DOE), Office of Basic Energy Sciences, Division of Materials Sciences and Engineering and performed, in part, at the Center for Integrated Nanotechnologies, an Office of Science User Facility operated for the US DOE Office of Science. Sandia National Laboratories is a multi-mission laboratory managed and operated by National Technology and Engineering Solutions of Sandia, LLC, a wholly owned subsidiary of Honeywell International, Inc., for the US DOE's National Nuclear Security Administration under contract no. DE-NA0003525. This Article describes objective technical results and analysis. Any subjective views or opinions that might be expressed in the paper do not necessarily represent the views of the US DOE or the United States Government. This study was funded by the US DOE Basic Energy Science Program (BES20017574) and the Materials Science Research Foundation LDRD program (230710).

## Author contributions

S.D., J.Y.T. and P.P.I. conceptualized the idea. P.P.I. designed and fabricated the metasurface. S.A. grew the epitaxial sample used to fabricate the device. P.P.I. and I.B. developed and built the ultrafast optical test setup. S.D. R.D. and P.P.I. developed the machine-learning framework. All authors contributed to the data analysis, writing and review process of the manuscript.

## Competing interests

The authors declare no competing interests
