## [Transparent Peer Review file · Nature Communications]

Self-driving lab discovers principles for steering spontaneous emission beyond conventional Fourier optics

Corresponding Author: Dr Prasad Iyer

Version 0:

Reviewer comments:

Reviewer #1

(Remarks to the Author)

The authors demonstrated a self-driving lab platform for designing reconfigurable active metasurfaces based on arrays of resonators with emitters, showcasing the ability to tune the emission over a wide field of view. Furthermore, the authors developed an equation learner to uncover the physical principles governing the steering of spontaneous emission, which reveals that both the first-order spatial derivative (grating-like) and the second-order spatial derivative (lens-like) of the index play important roles.

The self-driving lab presented in this manuscript efficiently controls the emission pattern, clearly surpassing conventional human intuition-based approaches. Moreover, the findings revealed by the equation learner are intriguing and will stimulate future research and discussions. Therefore, I recommend this manuscript for publication after minor revision. I have one major concern and several minor questions and suggestions.

Major Concern

In Figure 3, the latent space is characterized by four features z_i , where i in $\{1, 2, 3, 4\}$. However, in the Methods section, the latent space of the VAE is described as three-dimensional. Please ensure the description is correct and consistent throughout the manuscript.

Minor Questions and Suggestions

1. On page 2, you wrote:

"High dimensionality of inputs: We leverage the manifold hypothesis within the physical sciences, positing that high-dimensional experiments often lie on a low-dimensional manifold."

Readers from the metamaterial community who are unfamiliar with the manifold hypothesis may find this statement unclear. It would be helpful to describe the manifold hypothesis in the context of physical sciences. For instance: a description such as: "While a physical system may appear complex, requiring a large number of parameters to describe at first glance, there is often a simpler underlying mechanism that can be described with far fewer independent parameters." could convey the essence of the manifold hypothesis in the context of physics.

2. Building on point 1, it is possible to test the manifold hypothesis in this physical system (the metasurface with steerable emission) using manifold learning. I suggest the authors perform dimensionality reduction on the optical pump pattern (e.g., Figure 2c, right panel) for different angles (e.g., could be 7 classes of angle ranges, as in Figure 2b).

I estimate that, by using a method like t-SNE (see Roweis' Stochastic Neighbor Embedding and van der Maaten's Visualizing Data Using t-SNE for details) to reduce the dimensionality of these patterns from 3840 to a lower dimension, such as 3D or 2D, each class would be well-clustered, and all classes well-separated. Observing such well-clustered and well-separated low-dimensional representations for each class would provide strong evidence supporting the manifold hypothesis in this physical system.

3. In the supplementary material, specifically Figure S2, you wrote:

"The slopes of the patterns generated by the VAE (blue) are much broader than both the training set and the state-of-the-art sawtooth patterns, confirming that the VAE generates novel patterns beyond human intuition."

The slope of the pump pattern is broader than that of human-designed patterns. This indicates that the pattern contains many sharp (high dy/dx) features, which are common in automatically designed patterns and not necessarily unique to results from a VAE or other machine learning techniques. For instance, inverse-designed patterns often exhibit such features. However, sharp patterns pose challenges for experimental realization (which could be solved by additional regularizations in inverse design).

It would be beneficial for the authors to describe the sharpest feature that this experimental platform is capable of implementing.

4. On page 3, you wrote:

"However, predicting and controlling emission patterns from aperiodic refractive index modulation remains challenging due to a lack of suitable models and simulation tools."

Could you elaborate further on this difficulty or the complexity of such simulations, possibly in the supplementary material?

5. On page 3, you first mentioned "steering efficiency." A specific level of improvement, 67%, is provided before the quantitative definition of this term, which could lead to confusion. According to later paragraphs and the caption of Figure 2, "steering efficiency" appears to refer to peak directivity (defined in an equation on page 4). If this understanding is correct, please consider unifying the terminology to "peak directivity" to improve the readability of this manuscript.

Reviewer #2

(Remarks to the Author)

In the manuscript, 'Self-driving lab discovers principles for steering spontaneous emission' by S. Desai et al. physical equations are discovered using an AI system for steering spontaneous emission. The work is interesting, however, I am unsure that it is suitable for Nature Communications and may be suited to a more specialised journal. A few comments are given below:

The main claims in the abstract are above four-fold enhancement over a 72 degree field of view. However, this is a maximum value for a certain angle, not for all angles.

Somewhat interestingly, based on the previous comment and from intuition, +ive and -ive directions are generally just opposite. However, from Figure 2c, it appears that this is not the case for any of the patterns. Some description of what is going on would be great. Similarly for the peak improvement in Figure 2b, the +ive 30 bar is around 2, while the -ive shows almost no improvement. This seems unexpected, as if there is a preferential direction. This shouldn't be the case (from our intuition). Is there any explanation?

In addition to the previous comment, is there any reason why the emission angle of $\sim +10$ degrees would have the highest improvement?

In the discussion of the pruned network, where an equation is created, the authors write that 'repeating this process for other steering orders allows us to develop a general model'. So where is it? Furthermore, if this is the case, then could we then use these 'new insights' to design and fabricate (or at least simulate) a static metasurface (e.g. using silicon) with the additional curvature term added? We would expect then this uncovered relationship to provide us with a structure that produces a higher grating efficiency than a standard grating, confirming the new insights and claims.

Version 1:

Reviewer comments:

Reviewer #1

(Remarks to the Author)

The authors have thoroughly addressed all of my previous comments. I recommend the manuscript for publication.

(Remarks on code availability)

Reviewer #2

(Remarks to the Author)

The authors answered my comments well, but I still do not consider the work to be suitable for Nature Communications.

1) There are various examples of work that has used a lens and grating function in a metasurface to collimate and direct emission from resonators, so the results are somewhat expected.

2) Combining answers from comments of reviewer 1 and 2, the authors acknowledge the limitation in finding a local minima with their agent and therefore non-symmetric and somewhat unintuitive results (which would be a good thing if the AI found something new or unexpected), but also claim that the results are beyond human-designed patterns. This does not seem to align, since human-designed patterns are based on the physics and limitations of the system.

3) Additionally, in reply to reviewer 1 comment 3, the 'natural smoothing out' of the sharp features would lead to lots of sharp designs falling back to the same (diffraction limited) experimental result, which would ultimately bias the system and learning. This should be taken into consideration and developed more.

4) I am unconvinced by the t-SNE plot given in Figure S3. It does not show well-clustered and well-separated representations, and therefore does not provide evidence supporting the manifold hypothesis of the system. The latent

space is therefore not really structured or useful for sampling and iterating on.

(Remarks on code availability)

Reviewer #1 (Remarks to the Author):

The authors demonstrated a self-driving lab platform for designing reconfigurable active metasurfaces based on arrays of resonators with emitters, showcasing the ability to tune the emission over a wide field of view. Furthermore, the authors developed an equation learner to uncover the physical principles governing the steering of spontaneous emission, which reveals that both the first-order spatial derivative (grating-like) and the second-order spatial derivative (lens-like) of the index play important roles.

The self-driving lab presented in this manuscript efficiently controls the emission pattern, clearly surpassing conventional human intuition-based approaches. Moreover, the findings revealed by the equation learner are intriguing and will stimulate future research and discussions. Therefore, I recommend this manuscript for publication after minor revision. I have one major concern and several minor questions and suggestions.

Major Concern

In Figure 3, the latent space is characterized by four features z_i , where i in $\{1, 2, 3, 4\}$. However, in the Methods section, the latent space of the VAE is described as three-dimensional. Please ensure the description is correct and consistent throughout the manuscript.

Thank you for pointing out this error. The latent space is four dimensional, we have modified the Methods section description to now read as: **“In this work, we use a latent space representation of size four making the latent space a four-dimensional Gaussian distribution.”** (Page 13 Lines 9-10)

Minor Questions and Suggestions

1. On page 2, you wrote: "High dimensionality of inputs: We leverage the manifold hypothesis within the physical sciences, positing that high-dimensional experiments often lie on a low-dimensional manifold."

Readers from the metamaterial community who are unfamiliar with the manifold hypothesis may find this statement unclear. It would be helpful to describe the manifold hypothesis in the context of physical sciences. For instance: a description such as: "While a physical system may appear complex, requiring a large number of parameters to describe at first glance, there is often a simpler underlying mechanism that can be described with far fewer independent parameters." could convey the essence of the manifold hypothesis in the context of physics.

Thank you for the suggestion. We have rephrased the sentence to be – **“High dimensionality of inputs: We leverage the manifold hypothesis within the physical sciences which suggests that: A complex physical system requiring a large number of parameters at first glance, can be described with far fewer independent parameters.”** (Page 2 Lines 28-30). Here, we would like to note that the simplicity or the complexity of the underlying mechanism discovered through this process remains to be proven at this stage.

2. Building on point 1, it is possible to test the manifold hypothesis in this physical system (the metasurface with steerable emission) using manifold learning. I suggest the authors perform dimensionality reduction on the optical pump pattern (e.g., Figure 2c, right panel) for different angles (e.g., could be 7 classes of angle ranges, as in Figure 2b). I estimate that, by using a method like t-SNE (see Roweis' Stochastic Neighbor Embedding and van der Maaten's Visualizing Data Using t-SNE for details) to reduce the dimensionality of these patterns from 3840 to a lower dimension, such as 3D or 2D, each class would be well-clustered, and all classes well-separated. Observing such well-clustered and well-separated low-dimensional representations for each class would provide strong evidence supporting the manifold hypothesis in this physical system.

We note that well-separated and clustered pump patterns exist for each of these steering angles (Figure 3c,d,e) when the pump patterns are generated using $y = ax^2 + bx$ (a – curvature and b – gradient of pump pattern). The orange and purple regions in these plots are not overlapped and form unique combinations of (a,b) which maximizes the directivity of emission towards specific angles. Prior work from Ref 57, enables us to state that a single dimensional sub-space – grating order can steer the emission over 70° substantiating the manifold hypothesis. Here, we are utilizing the VAE to generate and explore patterns beyond the one-dimensional sub-space defined by the grating order.

As suggested by the reviewer, we also performed a t-SNE to reduce pump patterns to a lower dimension.

Figure S3: Latent space visualizations of families of pump patterns. Two-component latent space visualizations of grating order (saw-tooth) patterns (blue), patterns defined as polynomials (orange), and some patterns generated by sampling the VAE latent space (green), where the low-dimensional latent space descriptions are obtained by performing a t-SNE analysis on the pump patterns. We see that families of high-dimensional pump patterns lie on low-dimensional manifolds, supporting our leverage of the manifold hypothesis to generate candidate experiments.

We have added this figure and the explanation below in the supplementary information section S2 (*Supplementary Information Page 5 Lines 14-20 and Figure S3*). “We perform a t-SNE to discover that the pump patterns can be reduced to 2 dimensions accurately. From Figure S3– we observe that the set of points described by the grating order (blue) forms a subset of the points described by polynomial pump patterns (orange) which in turn forms a subset of the patterns generated by the VAE (green), as expected. We find that families of pump patterns lie on low dimensional manifolds that effectively describe their structural features. Iterating over these complex low-dimensional features in pump patterns directly relate to directivity, and our optimization method (active learning) samples the low-dimensional space to discovers patterns with high directivity.”

We have also modified the main manuscript to point to this information. “We also illustrate that the traditional grating order-based patterns and polynomial patterns lie on a subset of the space of patterns generated by sampling the VAE’s latent space (See Supplementary Information S2). This analysis confirms the validity of the manifold hypothesis in our problem, such that sampling this low dimensional space is sufficient to discover governing equations relating features of high-dimensional pump patterns to the observed directivity.” (*Page 6 Lines 3-8*). We greatly appreciate this insightful suggestion, as it helps to clarify the manifold hypothesis's applicability and enhances the readers' understanding of our approach.

3. In the supplementary material, specifically Figure S2, you wrote: "The slopes of the patterns generated by the VAE (blue) are much broader than both the training set and the state-of-the-art sawtooth patterns, confirming that the VAE generates novel patterns beyond human intuition." The slope of the pump pattern is broader than that of human-designed patterns. This indicates that the pattern contains many sharp (high dy/dx) features, which are common in automatically designed patterns and not necessarily unique to results from a VAE or other machine learning techniques. For instance, inverse-designed patterns often exhibit such features. However, sharp patterns pose challenges for experimental realization (which could be solved by additional regularizations in inverse design).

It would be beneficial for the authors to describe the sharpest feature that this experimental platform is capable of implementing.

We thank the reviewer for this comment, and agree with them on the observation that automatically generated patterns can often result in artificial features not realizable in experiments. Fortunately, our experimental setup inherently smooths such sharp features due to the natural propagation of electromagnetic fields. Since the active learning agent operates on experimental feedback, optimal patterns with sharp local slopes will be effectively converted to their “smoothed out” versions. Thus, while the VAE generates patterns with a wide range of slopes, patterns with artificially high local slopes will not affect our experimental measurements, and thus the optimization to find optimal patterns. We have added statements to supplementary information section S2, highlighting the sharpest feature possible for our optical telescope imaging the spatial light modulator to the metasurface. “Note that the VAE can generate pump patterns with large local slopes (over a few pixels). However, the sharpest feature realizable on the SLM is 30 pixels (3 microns) equivalent

to 1% of the length of the pump pattern. This feature length is defined by the diffraction limit of the telescope imaging the SLM image to the metasurface at 800 nm. The sharpest patterns (high dy/dx) generated by the VAE which are beyond the capabilities of the experimental setup (due to diffraction limit) would be smoothed (blurred) out naturally by the propagation of the electromagnetic fields. In these scenarios where the VAE pattern generates high spatial gradients, the active learning agent can only receive the experimental feedback with smoothed out pump patterns, thus predicting optimal pump patterns that account for these aberrations.” (*Supplementary Information Page 5 Lines 5-13*).

4. On page 3, you wrote:

"However, predicting and controlling emission patterns from aperiodic refractive index modulation remains challenging due to a lack of suitable models and simulation tools." Could you elaborate further on this difficulty or the complexity of such simulations, possibly in the supplementary material?

Yes, we have expanded the supplementary material to include a commentary on the current state-of-the-art in simulating spontaneous (incoherent) emission from metasurfaces, and modified the manuscript to point to this information (*Page 3 Line 25*).

“Simulating incoherent emission from metasurfaces: Conventional Finite-Difference Time-Domain (FDTD) simulations that are used to model the behavior of metasurfaces rely on the assumption of coherent sources (with periodic boundary conditions), which limits interactions to coherent scattering processes. Point dipole sources, placed within resonators, enable us to predict the local density of photonic states and the Purcell enhancement for a single dipole. However, adding multiple dipoles within the same simulation forces them to be coherent with respect to each other. Therefore, to model incoherent emission from a metasurface with a high density of spatially separated, mutually incoherent embedded quantum dot emitters, we need to perform independent FDTD simulations for each point dipole source. Within our metasurface, each nanopillar has 5 layers of high-(spatial) density InAs quantum dots (QDs) epitaxially grown with each layer having at least 30-40 independent quantum emitters – resulting in ~175-200 QD per nanopillar. With our metasurface consisting of 750x750 (~500k), we estimate a total of ~ 98 million QDs which need to be independently simulated for each spatial index pattern projected onto the metasurface. This is a significant challenge, and we thus use closed-loop experimental feedback to discover governing equations, instead of relying on simulations.” (*Supplementary Information Page 3 Lines 21-31 and Page 4 Lines 1-4*).

5. On page 3, you first mentioned "steering efficiency." A specific level of improvement, 67%, is provided before the quantitative definition of this term, which could lead to confusion. According to later paragraphs and the caption of Figure 2, "steering efficiency" appears to refer to peak directivity (defined in an equation on page 4). If this understanding is correct, please consider unifying the terminology to "peak directivity" to improve the readability of this manuscript.

Thank you for pointing this out, we have now defined steering efficiency to be the directivity, which peaks at 67% - now referred to as the peak directivity, making the appropriate changes in the manuscript. (*Page 4 Line 1, Page 7 Figure 2 caption*)

Reviewer #2 (Remarks to the Author):

In the manuscript, ‘Self-driving lab discovers principles for steering spontaneous emission’ by S. Desai et al. physical equations are discovered using an AI system for steering spontaneous emission. The work is interesting, however, I am unsure that it is suitable for Nature Communications and may be suited to a more specialised journal. A few comments are given below:

We thank the reviewer for their thoughtful evaluation and their meaningful comments which we believe has enabled us to develop a deeper physical understanding our results and helped us clarify the significance of our work to a broad interdisciplinary audience. This work demonstrates the first autonomous, AI-driven laboratory that discovers interpretable physical principles governing light-matter interactions—unlike "black box" ML approaches, our system generates human-interpretable equations revealing fundamental physics, establishing a new paradigm for autonomous scientific research applicable across physics, chemistry, materials science, and beyond. The research bridges AI, nanophotonics, materials science, and autonomous systems—precisely the interdisciplinary work Nature Communications champions, with our domain-agnostic framework addressing the broader challenge of accelerating discovery in high-dimensional parameter spaces. Beyond methodological innovation, we report new principles for steering spontaneous emission using reconfigurable metasurfaces, representing a fundamental advance in light-matter interactions with implications extending far beyond specialized photonics applications. The discovered principles enable transformative applications from next-generation LEDs and photodetectors to AR/VR displays, LIDAR systems, and Li-Fi communications, demonstrating technological relevance across multiple industries. The manuscript presents two significant contributions: (1) a generalizable framework for AI-driven autonomous scientific discovery, and (2) fundamental insights into controlling incoherent light emission. This pioneering integration of autonomous experimentation, interpretable AI, and physics discovery represents a new frontier in scientific methodology suitable for Nature Communications' broad, interdisciplinary readership.

The main claims in the abstract are above four-fold enhancement over a 72 degree field of view. However, this is a maximum value for a certain angle, not for all angles.

Thank you for pointing this out, you are correct. We have changed this to “**up to a four-fold enhancement ...**” (*Page 1 Line 20*)

Somewhat interestingly, based on the previous comment and from intuition, +ive and -ive directions are generally just opposite. However, from Figure 2c, it appears that this is not the case for any of the patterns. Some description of what is going on would be great. Similarly for the peak

improvement in Figure 2b, the +ive 30 bar is around 2, while the -ive shows almost no improvement. This seems unexpected, as if there is a preferential direction. This shouldn't be the case (from our intuition). Is there any explanation?

Thank you for this insightful comment. The optimal patterns in Figure 2c represent the local-maxima of the active learning algorithm, optimizing over the latent space of the variational autoencoder after running for a fixed budget of 300 experiments (of which 100 are used as training data). We understand and appreciate the reviewer's intuitive expectation for symmetrical results. However, due to the high-dimensional nature and limited number of iterations within our experimental budget, the current optimization scheme naturally finds local maxima rather than global ones. We explicitly note this limitation in the revised manuscript. We have demonstrated that through our prior publications (see ref 57, figure 4) and systematic variations in the pump patterns ($y = ax^2+bx$) in this work, see Figure 3 and Figure S6, that we achieve symmetric directivities, partially lining up with our expectations. We have highlighted this point in the manuscript: “This loop continues until an optimum is reached, or a pre-set experimental budget is reached. Here the active learning agent is limited by our experimental budget and we note that maxima in directivity (figure of merit) achieved represents local maxima, and not a global maximum of the optimization problem”. (Page 6 Lines 13-16)

In addition to the previous comment, is there any reason why the emission angle of $\sim+10$ degrees would have the highest improvement?

Thank you for this question. Following the previous argument of achieving local maxima using the active learning agent, we cannot currently assign specific reasons for large improvements for these specific angles. Future systematic investigations are necessary to thoroughly understand this directional asymmetry, which we acknowledge as an important area for further exploration. However, we see that large improvements to the grating-order like pump patterns exist across all angles, indicating that despite our optimizer suggesting local maxima, we find significant improvements, and have discovered a new way to steer light which has a much higher directivity of emission (efficiency) over all steering angles measured within the field of view. Performing high-dimensional optimizations which respect the underlying physics (e.g. symmetries) is a significant challenge in material science and chemistry, and perhaps future work can address this by designing a more customized active learning based optimization scheme which respects the underlying symmetries of this problem.

In the discussion of the pruned network, where an equation is created, the authors write that ‘repeating this process for other steering orders allows us to develop a general model’. So where is it? Furthermore, if this is the case, then could we then use these ‘new insights’ to design and fabricate (or at least simulate) a static metasurface (e.g. using silicon) with the additional curvature term added? We would expect then this uncovered relationship to provide us with a structure that produces a higher grating efficiency than a standard grating, confirming the new insights and claims.

Thank you for pointing this out. In the supplementary material (section S3), we have included the following equations that we discover for various emission angles, demonstrating the strong

dependency of directivity on both the grating order (slope) and lens-like (curvature) pump pattern characteristics. (*Supplementary Information Page 7 Lines 1-8*)

$$D_g(+14^\circ) = -0.075a + 0.03b - 0.69(0.74 \sin(x) - \sin(y))^2 + 0.055(-b + 0.3 \sin(y))^2 \\ - 0.33(0.02a + 0.13b + \sin(x) - 0.67 \sin(y))^2 + 0.1(\sin(x) - \sin(y)) \\ + 0.77$$

where $x = 1.85 a + 0.53 b$ & $y = 2.09 a + 0.15 b$

$$D_g(-14^\circ) = -0.02b - 0.26(0.55a - b + 0.152 \sin(x))^2 + 0.054(a - 0.87b + 0.24 \sin(x))^2 \\ + 0.23(0.53a - b + 0.01 \sin(x) - 0.65 \sin(y))^2 + 1.11$$

where $x = 2.16 a + 1.2 b$ & $y = 1.1 a - 0.86 b$

These equations represent optimal fits to our extensive experimental dataset (encompassing 84,000 results from Figure 3 and Supplementary Figure S6), which can be considered as projections of a ‘master governing equation’ within our measurement space. While our dataset provides substantial density with respect to curvature (a) and gradient (b) parameters of the pump-pattern, we acknowledge that it remains relatively sparse across the steering angle dimension. Additional experimental investigation would be necessary to develop a more comprehensive understanding across the full angular space. Nevertheless, by analyzing the existing equations collectively, we have developed an initial intuitive understanding of the relationship between parameters a and b—specifically examining how the ratio a/b must be adjusted to optimize light steering toward specific angles. We have incorporated these equations into the supplementary section and refined the main text to note: " Examples of such additional equations are provided in the Supplementary Information Section S3. Analyzing these equations, we can observe a consistent trend wherein the optimal a/b ratio for maximizing light emission gradually increases as the redirection angle changes from -36° to $+36^\circ$." (*Page 11 Lines 13-17*)

As the reviewer, has correctly identified, these spatial index profiles could potentially enable the development of static metasurfaces capable of re-directing the emission from within the structure itself. However, we anticipate that such static designs would be most applicable to systems similar to our specific metasurface configuration (high-index resonators on reflective distributed Bragg grating substrates). It is worth noting that comparable spatial phase profiles have previously been employed to create static metasurface designs intended to focus photoluminescence generated within the metasurface. [Ref 56] We thank the reviewer for highlighting this promising direction. Indeed, translating these insights into static device designs is a compelling future research direction, and we are actively exploring this in a separate initiative to do justice to its complexity and potential.

Reviewer #1 (Remarks to the Author):

The authors have thoroughly addressed all of my previous comments. I recommend the manuscript for publication.

We thank the reviewer for their comments and careful consideration of our manuscript.

Reviewer #2 (Remarks to the Author):

The authors answered my comments well, but I still do not consider the work to be suitable for Nature Communications.

We are grateful to the reviewer for their careful reading of the manuscript and the follow up comments. The *central novelty* in the manuscript stems from our self-driving lab architecture where the combination of a generative model (for experiment generation), active learning (design of experiments) and equation learner networks (interpreting the results) enables us to efficiently sample high-dimensional design spaces increasing the probability of discovering novel interpretable scientific principles. Consequently, we were able to discover a new way to steer incoherent light going beyond the governing principles of Fourier optics. We have given detailed explanations supporting the same as answers to the questions below and reflected those changes in the manuscript to highlight the novelty in our work.

1) There are various examples of work that has used a lens and grating function in a metasurface to collimate and direct emission from resonators, so the results are somewhat expected.

We agree with the reviewer that prior example of using lens and grating functions to direct emission from metasurfaces exist in literature, and we have already cited them in our work as well, see ref 56 in the main manuscript. The results in the ref 56, demonstrates a method to focus the light emission from a metasurface as opposed to dynamically steering the emission here. However, we would like to point out that all of those examples strictly follow design principles based on Fourier transforms – i.e. a spatial phase profile on the metasurface gets transformed into a far-field momentum space profile of light emission. Our results demonstrated in our manuscript – as discovered by our self-driving lab framework – deviates from this fundamental principle of gradient metasurface design. Our key result suggests that for steering incoherent emission from within the metasurfaces goes beyond the principles of Fourier transform – which has not been demonstrated for any nanophotonic or optical device to the best of our knowledge.

Let's consider Figure 3 c,d,e from the manuscript, repeated below for ease of reading: Each of these color maps represents the directivity of emission from the metasurface towards -14.4° , 0° , and $+14.4^\circ$ degrees while collecting the emission over 74° field of view. Each point on this color map represents the directivity from the metasurface for an optical pump pattern defined as $y = ax^2 + bx$ – where a is varied along the x -axis from -40 to $+40$

and b is varied along the y -axis from -30 to $+30$ covering both concave ($a < 0$) and convex ($a > 0$) lenses, and negative ($b < 0$) and positive ($b > 0$) gratings.

Focusing on each of these figures on a quadrant-by-quadrant basis, we observe that only a combination of convex lens and a positive grating, and a concave lens and a negative grating, show signatures of high directivity. This has been verified experimentally for all measured angles from -36.6° to $+36.6^\circ$. Simultaneously, directivity of emission for a pump pattern formed as a combination of a convex lens and a negative grating, and a concave lens and a positive grating does not steer light for any emission angle. If we follow conventional principles of Fourier transforms, and a convex lens and a positive grating steers light towards positive angles, then a convex lens and a negative grating should steer towards negative angles. We do not observe that for any of our measured angles. Similarly, for a system steering light using a positive lens and a positive grating phase pattern, a negative lens and a negative grating should not be able to steer light into the same direction. In all of our measured angles, we observe that this is also violated.

The results presented here represent a new principle of steering incoherent light emission from reconfigurable metasurfaces, which goes beyond conventional principles Fourier Optics. We believe this represents a fundamental difference from all existing demonstration of steering (both coherent and incoherent) light in general. Moreover, the primary novelty of the paper as whole stems from the fact that a closed-loop, self-driving lab framework led us to this new method to steer light which goes beyond our human intuition governed by Fourier Optical principles.

We have further highlighted this as two major changes:

- a) We have updated the title: “Self-driving lab discovers principles for steering spontaneous emission beyond conventional Fourier optics”
- b) We have added the following statements to the introduction: “Using the results from our self-driving lab, we discovered a novel approach to steer incoherent emission – going beyond conventional Fourier optical principles that are based solely on momentum matching of light. We demonstrate that a spatial index profile that is a combination of a positive (convex) lens and positive (sawtooth) grating, when imposed on the metasurface, can steer light emission across 74° field of view. Similarly, a negative (concave) lens and a negative (sawtooth) grating also steer emission across the same field of view, while other spatial index combinations (e.g. positive (negative) lens with a negative (positive) grating) do not steer the emission from our reconfigurable light-emitting metasurface.” see Page 4 Lines 5-13.

We also note that a paragraph just before the conclusion (see *Page 12 Lines 13-31 and Page 13 Lines 1-2*) explains these results in detail, demonstrating how we are going beyond conventional Fourier optics.

2) Combining answers from comments of reviewer 1 and 2, the authors acknowledge the limitation in finding a local minima with their agent and therefore non-symmetric and somewhat unintuitive results (which would be a good thing if the AI found something new or unexpected), but also claim that the results are beyond human-designed patterns. This does not seem to align, since human-designed patterns are based on the physics and limitations of the system.

Thank you for highlighting this point of confusion. While we agree that humans (subject matter experts) could design patterns based on the physics, and symmetries, of the system, they might fail to generate complex patterns that could steer emission efficiently. We can see this in Figure S2, which quantifies the variety of patterns generated by the VAE, in terms of their local slopes (local momentum of emission by conventional design principles). We see that the patterns generated by the VAE (blue) have ~100x more variation in the local-slope than the state-of-the-art sawtooth patterns (green region) and the training dataset consisting of some basic polynomial-based patterns (orange region). That is, the patterns generated by the VAE are beyond human-design in their complexity. However, since the VAE does not incorporate any underlying physics, the patterns in this design space are not restricted/constrained based on underlying symmetries of the spatial index profiles. The active learning agent samples the latent space of this VAE with the aim to maximize the directivity of emission from the metasurface. The active learning agent is agnostic of the nature of the pump patterns, all of which are generated by the VAE. As we demonstrated in our manuscript, Figure 2, the local maxima discovered by the active learning agent are complex spatial index profiles with high directivity, and could not be designed based on symmetric arguments alone. Therefore, we believe that there need not exist any alignment between local-maxima patterns (function of the active learning agent) and characterization of pump patterns (symmetry based vs VAE). We have modified the main manuscript to highlight this point further: **“The patterns discovered by the AL agent resulting in high directivity, represent spatial index profiles beyond conventional optical elements, which are typically defined based on symmetries governed by the spatial phase profile of the device.”** (see *Page 8, line 24-26*)

3) Additionally, in reply to reviewer 1 comment 3, the 'natural smoothing out' of the sharp features would lead to lots of sharp designs falling back to the same (diffraction limited) experimental result, which would ultimately bias the system and learning. This should be taken into consideration and developed more.

We thank the reviewer for focusing on this point. As reviewer 1 previously pointed out, it is well known that generative models like VAE could generate patterns which are unaware of the physical limitation of the system. However, we overcome the intrinsic limitation of

generative model through closed-loop experimental feedback. We have modified the main manuscript to acknowledge this point: “While the VAE could generate patterns with high-spatial frequency going beyond the diffraction limit of the optical system, we note that the experimental setup imaging the pump pattern from the SLM to the metasurface will naturally smooth out these high-frequency elements.” (see Page 8, Lines 5-8)

We agree with the reviewer that there could be a bias in the system learning, where different points in a sub-space of the VAE latent space, which should represent different pump patterns, could result in similar directivity, since the experiment ‘smooths out’ the sharp features of the different pump patterns, making them similar to the active learning optimizer. However, given the extremely large design space of possible complex pump patterns, we still expect to find, and do find, novel patterns that have high directivity which do not have sharp spatial features. We have specifically pointed this out in the main manuscript:” Additionally, we note the patterns discovered by the AL agent do not have any high-spatial frequency features beyond the diffraction limit of the system. Thus, the AL agent...” (see Page 8, Lines 27-28)

Additionally, while we present the results from our active learning (Figure 2), we also follow it up with a brute-force parameter sweep based verification (Figure 3 c,d,e), where the pump patterns are far above the diffraction limit of the system. We also note that the optimal pump patterns for all our steered angles from -36.6° to $+36.6^\circ$ (Figure 2d) contain features that are significantly above the diffraction limit suggesting that the active learning agent is not getting biased with high-spatial frequency pump patterns that could be generated by the VAE.

4) I am unconvinced by the t-SNE plot given in Figure S3. It does not show well-clustered and well-separated representations, and therefore does not provide evidence supporting the manifold hypothesis of the system. The latent space is therefore not really structured or useful for sampling and iterating on.

We agree with the reviewer that the t-SNE plot does not show well separated clusters. However, that does not mean that it does not provide evidence supporting the manifold hypothesis of the system. The cluster representing the output of the VAE forms a superset of the patterns providing high-directivity, which is expected since the VAE generates pump patterns similar to, and beyond, its training set (see Figure S2). The training set was a series of pump patterns involving grating orders and polynomials (the other two clusters in the t-SNE plot).

The manifold hypothesis suggests that for high-dimensional systems within the physical sciences there exists a lower-dimensional sub-space which effectively captures the properties of the system. As mentioned previously, we have multiple experimental measurements that we believe supports our claim.

A) Figure 1b – we can steer the light emission using a single dimensional grating parameter – which represents a lower dimensional (single parameter) representation of a high-dimensional pump pattern (array of length 3840).

B) The fact that we are able to train a VAE to accurately reconstruct (and also generate) pump patterns using just a 4-dimensional latent space, also suggests that for the pump patterns of interest, there exist a 4-dimensional representation of these patterns that is sufficient, and that the manifold hypothesis is not violated.

C) Figure 3 – Our results suggest that a 2-dimensional representation of the pump pattern (a – spatial curvature, and b – spatial gradient) forms a superior latent-space for controlling the far-field emission pattern when compared with just a spatial gradient.